# Synthetic Sulfide Concentrate Dissolution Kinetics in HNO_3_ Media

**DOI:** 10.3390/ma15228149

**Published:** 2022-11-17

**Authors:** Oleg Dizer, Kirill Karimov, Aleksei Kritskii, Denis Rogozhnikov

**Affiliations:** Laboratory of Advanced Technologies in Non-Ferrous and Ferrous Metals Raw Materials Processing, Department of Non-Ferrous Metals Metallurgy, Ural Federal University, 620002 Yekaterinburg, Russia

**Keywords:** leaching, nitric acid, tennantite, chalcopyrite, sphalerite, pyrite, sulfide minerals, kinetics, ion concentration, catalyst, galvanic coupling, mechanism

## Abstract

The nature of tennantite (Cu_12_As_4_S_13_), chalcopyrite (CuFeS_2_) and sphalerite (ZnS) particles’ mixture dissolution in nitric acid (HNO_3_) media was investigated in this study. The effects of temperature (323–368 K), HNO_3_ (1–8 mol/L) and Fe^3+^ (0.009–0.036 mol/L) concentrations, reaction time (0–60 min) and pyrite (FeS_2_) additive (0.5/1–2/1; FeS_2_/sulf.conc.) on the conversion of the minerals were evaluated. It has been experimentally shown that the dissolution of the mixture under optimal conditions (>353 K; 6 mol/L HNO_3_; FeS_2_/synt. conc = 1/1) allows Cu_12_As_4_S_13_, CuFeS_2_ and ZnS conversion to exceed 90%. The shrinking core model (SCM) was applied for describing the kinetics of the conversion processes. The values of E_a_ were calculated as 28.8, 33.7 and 53.7 kJ/mol, respectively, for Cu_12_As_4_S_13_, CuFeS_2_ and ZnS. Orders of the reactions with respect to each reactant were calculated and the kinetic equations were derived to describe the dissolution rate of the minerals. It was found that the interaction between HNO_3_ solution and Cu_12_As_4_S_13_, CuFeS_2_ and ZnS under the conditions investigated in this are of a diffusion-controlled nature. Additionally, the roles of Fe(III) in the initial solution and FeS_2_ in the initial pulp as catalysts were studied. The results indicated that the increase in Fe^3+^ concentration significantly accelerates the dissolution of the mixture, while the addition of FeS_2_ forms a galvanic coupling between FeS_2_, and Cu_12_As_4_S_13_ and CuFeS_2_, which also accelerates the reaction rate. The results of the study are considered useful in developing a hydrometallurgical process for polymetallic sulfide raw materials treatment.

## 1. Introduction

Intensive exploitation of the primary raw materials in the non-ferrous metals industry resulted in the depletion of the rich deposits as a consequence of seeking alternative sources of raw materials. In view of the current trend in decreasing the metal content in the ore, smelters adapt to lower grade and technogenic raw materials [1]. One of the most specific raw material types are arsenic-containing ores and concentrates since their treatment is associated with additional environmental risks and technological complexity. Thus, along with the familiar minerals in copper industry such as chalcopyrite (CuFeS_2_), covellite (CuS), chalcocite (Cu_2_S), bornite (Cu_5_FeS_4_) and sphalerite (ZnS), sulfide ores occasionally contain minerals of the Fachlor group, such as tennantite (Cu_12_As_4_S_13_) and tetrahedrite (Cu_12_Sb_4_S_13_), which complicates the processing of the obtained ore/concentrates using typical approaches [2].

Pyrometallurgical processing of the arsenic-containing raw materials admits releasing arsenic dust and other waste into the atmosphere, which poses a significant danger to humans and the environment [3,4,5,6,7,8,9,10,11,12]. Therefore, pyrometallurgical processing of such materials is considered unacceptable today, and hence, undergoes strict government regulation in the most countries.

As an alternative to conventional smelting, various hydrometallurgical approaches are currently being researched, including the following: acid [13,14,15] and ammonia leaching [16], alkaline [17,18,19] and autoclave approaches [20,21,22], bioleaching [23,24], etc. Although autoclave approaches have been known since the middle of the last century and have found success and wide application for gold–silver sulfide raw materials treatment [25], atmospheric systems are also of great interest in hydrometallurgy [26].

One of the most promising approaches in this direction is an application of nitric acid (HNO_3_) as a reactant for atmospheric leaching, since HNO_3_ allows the most complete decomposition of the sulfide matrix [27,28]. Thus, HNO_3_ leaching under optimal conditions could provide both the transference of valuable metals into solution and arsenic and other toxic elements into stable and environmentally friendly compounds [29,30,31,32,33].

This study is aimed at evaluating the fundamental effectiveness of sulfide low-grade raw-materials treatment using HNO_3_ as the reactant for atmospheric leaching and also at evaluating the kinetic characteristics of the dissolution processes. High-purity specimens of Cu_12_As_4_S_13_, CuFeS_2_ and ZnS were used in this study to create the mixture of sulfides (synthetic concentrate), which was subjected dissolution in HNO_3_ media. The choice of these minerals was for the purpose of creating the most refractory raw materials through hydrometallurgical treatment.

The shrinking core model (SCM) was applied for describing the kinetics of the dissolution processes. The effects of temperature, reaction time and HNO_3_ concentration on the conversion of the minerals were evaluated. We also evaluated the effect of the pyrite (FeS_2_) additive and the Fe^3+^ concentration in the initial solution, since a number of studies reported [34,35,36,37,38,39,40,41,42,43,44] using the mentioned reactants as catalysts.

The results of the study are considered useful in developing hydrometallurgical processes for polymetallic sulfide raw-materials treatment.

## 2. Materials and Methods

### 2.1. Materials

High-purity specimens of Cu_12_As_4_S_13_, CuFeS_2_, ZnS and FeS_2_ originating from the Uchalinskii deposit (Sverdlovsk region, Russia), the Vorontsovskii deposit (North Ural region, Russia), the Karabashskii deposit (South Ural region, Russia) and the Berezovskii deposit (Sverdlovsk region, Russia), respectively, were used in this study (Figure 1A–D). The samples for experiments were obtained from the ground crystals by wet sieving (20–40 µm), following the mixing of Cu_12_As_4_S_13_, CuFeS_2_ and ZnS in the proportion of 1/0.36/0.17 (by weight). The obtained mixture represented the sulfide concentrate of the complex composition (Table 1). The established ratio of the minerals in synthetic concentrate is typical for the industrial Cu-As concentrate of the Uchalinskii deposit [44]. An additive of FeS_2_ into the leaching pulp was used in the study to evaluate the galvanic effect between FeS_2_, Cu_12_As_4_S_13_ and CuFeS_2_.

### 2.2. Experimental Procedure

The dissolution of the synthetic concentrate in HNO_3_ solution was conducted in a 500 mL round-bottom borosilicate glass reaction vessel (Lenz Laborglas GmbH & Co. KG, Wertheim, Germany) with a thermostatic jacket, which was thermostated using a Huber KISS-205B circulator (Huber Kältemaschinenbau AG, Offenburg, Germany). The reactor is equipped with an IKA EUROSTAR 20 digital overhead stirrer (IKA^®^-Werke GmbH & Co. KG, Staufen, Germany).

The values of the change in the Gibbs energy were calculated using the HSC Chemistry Software v. 9.9 (Metso Outotec Finland Oy, Tampere, Finland).

To examine the synthetic concentrate dissolution in HNO_3_ solution, a 5 g synthetic concentrate sample (particle size +20–40 µm) was added into the 200 mL of 1–8 mol/L HNO_3_ solution when the temperature inside the leaching vessel reached the desired temperature (323–368 K). When applicable, an additive of FeS_2_ was added to the initial pulp with different mass ratios of FeS_2_ to synthetic concentrate sample: 0.5/1, 1/1, 1.5/1, 2/1. The latter was equal to 2.5, 5, 7.5 and 10 g of FeS_2_ additive, respectively. During the experimental runs, samples were withdrawn from the reaction vessel at regular time intervals (1, 1.5, 2, 5, 15, 30, 45 and 60 min) using an automatic dispenser Sartorius Proline (Mine-beaIntecAachen GmbH & Co. KG, Aachen, Germany) and filtered using a 45 µm syringe filter. The final pulp after the experiments was filtered using a Buchner funnel with filter paper. The solid residue (cake) was washed with distilled water, dried in an oven at a temperature of 274 K at least for 1 h, weighed, and analyzed further for the residue characterization. The solution samples received during the experiments as well as solutions after filtration of the final pulp were subjected to volume measuring and analysis for Cu, Fe, As and Zn. The received data was used in calculating the fraction reacted (X) and the conversion (E, %) of the minerals with the following equations:(1)$$X=\frac{m_{s}}{m_{i}}$$
(2)$$E=\;\frac{m_{s}}{m_{i}}\cdot 100\%$$
where m_s_ and m_i_ are the mass of As, Cu and Zn in solution after the treatment and the mass of As, Cu, Zn in the initial synthetic concentrate, respectively.

### 2.3. Analysis

The mineralogical and chemical compositions of the minerals and solid residues were determined based on wave dispersive spectrometry (ARL Advant’X 4200, Thermo Fisher Scientific, Walthamm, MA, USA), X-ray diffraction “XRD” (XRD-7000, Shimadzu, Kyoto, Japan), scanning electron microscopy “SEM” (JSM-6390LV, Jeol, Tokyo, Japan) with a JED 2300 Energy Dispersive X-ray Analyzer (EDX) (Jeol, Tokyo, Japan), wet analysis using inductively coupled plasma mass-spectrometry “ICP-MS” (Elan 9000, PerkinElmer, Waltham, MA, USA) and laser diffraction (Helos/BR, Sympatec, Clausthal-Zellerfeld, Germany). The solid materials were ground in a planetary mill (Pulverisette 6, Fritsch GmbH & Co. KG, Welden Germany) and dissolved in an aqua regia before ICP-MS analysis. The sulfur content was analyzed using a carbon/sulfur analyzer (CS 230, LECO, St. Joseph, MI, USA). Solution samples were analyzed by ICP-MS.

## 3. Results and Discussion

### 3.1. Effect of Temperature

Temperature range under investigation in this study was taken as 323–368 K, since previously published works [45] show a low conversion of sulfide minerals at ambient temperature. Thus, the chosen temperature range is of primary interest for the atmospheric leaching processes investigation.

Increasing the temperature seemed to have a significant effect on the conversion of Cu_12_As_4_S_13_ (Figure 2a), CuFeS_2_ (Figure 2b) and ZnS (Figure 2c). To illustrate, at 323 K for 60 min, only 65% of Cu_12_As_4_S_13_, 45% of CuFeS_2_ and 53% of ZnS were reacted, while at 368 K conversion increased to 96, 75 and 98%, respectively for Cu_12_As_4_S_13_, CuFeS_2_ and ZnS.

According to the conversion profiles, the progress of the reaction was observed to slow down with time. Additionally, the presence of a clear inflection line in curves allowed us to suggest an internal diffusion mechanism of the interactions. The possible internal diffusion may cause elemental sulfur formation during the processing of the synthetic concentrate by the following reactions:CuFeS_2_ + 16HNO_3_ = FeSO_4_ + CuSO_4_ + 16NO_2_+ 8H_2_O; ΔG^0^_353_ = −1187 kJ/mol,(3)
CuFeS_2_ + 10HNO_3_ = Fe(NO_3_)_3_ + Cu(NO_3_)_2_ +2S^0^ + 5NO_2_ + 5H_2_O; ΔG^0^_353_ = −438 kJ/mol,(4)
Cu_12_As_4_S_13_ + 64HNO_3_ = 12Cu(NO_3_)_2_ + 4H_3_AsO_4_ + 13H_2_SO_4_ + 40NO + 13H_2_O; ΔG^0^_353_ = −1866 kJ/mol,(5)
Cu_12_As_4_S_13_ + 38HNO_3_ = 12Cu(NO_3_)_2_ + 4H_3_AsO_4_ + 13S^0^ + 14NO + 13H_2_O; ΔG^0^_353_ = −762 kJ/mol,(6)
ZnS + 8HNO_3_ = ZnSO_4_ + H_2_SO_4_ + 8NO_2_ + 4H_2_O; ΔG^0^_353_ = −640 kJ/mol(7)

### 3.2. Effect of HNO_3_ Concentration

Figure 3 illustrates the effect of increasing the initial concentration of HNO_3_ ranging from 1 to 8 mol/L on the conversion of Cu_12_As_4_S_13_, CuFeS_2_ and ZnS. The increase in HNO_3_ concentration seemed to also improve the rate and extent of the sulfides conversion. Over the reaction period, the conversion of Cu_12_As_4_S_13_ (Figure 3a), CuFeS_2_ (Figure 3b) and ZnS (Figure 3c) significantly increased from 35, 24 and 67%, respectively, at 1 mol/L HNO_3_ in solution to 94, 81 and 98% at 8 mol/L HNO_3_.

The significant effect of HNO_3_ concentration on the reaction rate and conversion extent may also indicate that the reactions are controlled by diffusion through the product layer, where the increasing HNO_3_ concentration in the initial solution leads to acceleration of S° to SO_4_^2−^ transformation and as consequence, the predominant course of reactions by Equations (3), (5) and (7).

### 3.3. Effect of Fe(III) Concentration

The effect of Fe (III) concentration ranging from 0.009 to 0.036 mol/L on the conversion of Cu_12_As_4_S_13_, CuFeS_2_ and ZnS was investigated. The results in Figure 4 show a moderate increase in the reaction rate by increasing the Fe (III) concentration. After 60 min of reaction at 0.009 mol/L Fe (III), 79, 65 and 91% of Cu_12_As_4_S_13_ (Figure 4a), CuFeS_2_ (Figure 4b) and ZnS (Figure 4c), respectively, were converted, compared with 90, 96 and 98% conversions, respectively at 0.036 mol/L. The interaction of main minerals in synthetic concentrate with Fe(III) are supposed to proceed according to the following reactions:CuFeS_2_ + 2Fe_2_(SO_4_)_3_ = CuSO_4_ + 5FeSO_4_ + 2S^0^; ΔG^0^_353_ = −65 kJ/mol,(8)
ZnS + Fe_2_(SO_4_)_3_ = ZnSO_4_ + 2FeSO_4_ + S^0^; ΔG^0^_353_ = −56 kJ/mol(9)
Cu_12_As_4_S_13_ + 13.5Fe_2_(SO_4_)_3_ + 6H_2_O = 12CuSO_4_ + 27FeSO_4_ + 14.5S^0^ + 4H_3_AsO_3_; ΔG^0^_353_ = −28 kJ/mol(10)

### 3.4. Effect of FeS_2_ Additive

Four different mass ratios of FeS_2_ to synthetic concentrate (0.5/1, 1/1, 1.5/1, 2/1, which is equal to 2.5, 5, 7.5 and 10 g of FeS_2_ adding) were used in the experiments to examine the effect of galvanic coupling. The results are shown in Figure 5.

As expected, increasing the mass of additive resulted in moderately improved conversion [35]. The obtained results indicate that the oxidation of Cu_12_As_4_S_13_ and CuFeS_2_ (Equations (4) and (6)) may proceed with the formation of S°, which passivates the surface of the minerals. At the same time, FeS_2_ may act as an alternative catalytic surface for these minerals. The latter provides the reduction of HNO_3_ on FeS_2_ surface and decomposition of other sulfide minerals [45].

In the experiments with 0.5/1 mass ratio, the conversion of Cu_12_As_4_S_13_ (Figure 5a), CuFeS_2_ (Figure 5b) and ZnS (Figure 5c) appeared to be limited to 64, 58 and 89%, respectively. Usage of the higher mass ratio (2/1) resulting the conversion of the minerals to be increased up to 83, 83 and 98%, respectively for Cu_12_As_4_S_13_, CuFeS_2_ and ZnS.

## 4. Characterization of Residues

### Characteristics of the Received Cakes

Figure 6 shows the SEM images (Figure 6a,b) and EDX-mapping of the residue (Figure 6c–f) after leaching the synthetic concentrate in HNO_3_ solution. EDX-mapping images confirm the formation of S° layer on the surface of unreacted synthetic concentrate particles. Thus, the dark layer over the muted points of Fe (Figure 6c) and Cu (Figure 6d) as well as bright green points on the image of joint EDX-mapping (Figure 6f) over all other components allow us to confirm the S° presence. Thus, the mentioned conditions suggest proceeding interactions by Equations (4) and (6).

The S° content in the residue was observed to be 56% and the conversion of sulfide sulfur to S° appeared to be about 38%. Under these conditions, the conversion of Cu_12_As_4_S_13_, CuFeS_2_, and ZnS was 59, 60 and 84%, respectively.

In contrast to the experiment without the FeS_2_ additive (Figure 6), the results with the additive (FeS_2_/synt. conc = 1/1) showed a lesser S° content.

The Fe and Cu points indicated in one component EDX-mapping images (Figure 7c,d, respectively) became brighter. According to Figure 7a,b, the residue has a heterogeneous surface resembling conglomerates, while the experiment without FeS_2_ additive shows a more homogeneous structure due to the covering of the particles by S°. The green zones in Figure 7f correspond to the distribution of S°, while the mixture of red and blue zones are copper minerals (Cu_12_As_4_S_13_ and CuFeS_2_) and FeS_2_.

Therefore, S° covers the surface of the synthetic concentrate in lesser extent, which confirms by the SEM-EDX residue investigation as well as the chemical composition of the residue—sulfide sulfur to S° transformation decreased to 23%, while the S° content in the solid residue decreased to 14%. Under these conditions of dissolution, the conversion of Cu_12_As_4_S_13_, CuFeS_2_ and ZnS was 87, 91 and 98%, respectively.

Figure 8 shows the XRD patterns of the solid residues after the dissolution of synthetic concentrate in HNO_3_ solution. The obtained data additionally confirms that the presence of FeS_2_ allows to limit the formation of S°.

SEM images (Figure 9) of the material after dissolution for 15 min (bend point in Figure 5 for Cu_12_As_4_S_13_ and CuFeS_2_) coupled with EDX analysis (Table 2) suggest that most of the S° was formed towards the end of this period and the subsequent dissolution of the material occurs at its coating by S°.

Therefore, it is appropriate to conclude that the diffusion in the system is the result of S° formation during the first 10–20 min of the experiment. After that, the dissolution process shifts to diffusion control.

## 5. Kinetics Analysis

As it was shown, the conversion of sulfides is significantly affected by temperature, HNO_3_ concentration and the presence of FeS_2_ in the system; that could mean possible control of the reactions by both chemical reaction and diffusion. To determine the limiting stage of the processes, the most commonly used kinetic equations of SCM describing liquid-solid reactions [46] were used to fit into the experimental data (Table 3). According to the results present in Figure 2, Figure 3, Figure 4 and Figure 5, the higher conversion degree during the initial period of reaction was observed for ZnS, therefore, kinetic analysis of the mineral was carried out in the period from 0 to 2 min, while for Cu_12_As_4_S_13_ and CuFeS_2_, from 0 to 60 min.

As shown in Figure 10, the SCM equation typically applied for diffusion kinetic system (Table 4, Equation (1)) can be used to describe the conversion processes with high values of the determination coefficient (R^2^).

The activation energy values (E_a_) were calculated using the Arrhenius law (Figure 11). Thus, E_a_ was determined as 28.8 kJ/mol for Cu_12_As_4_S_13_ and 33.7 kJ/mol for CuFeS_2_, values that are typical for inner-diffusion processes [46]. The activation energy values for ZnS treatment were determined as 53.7 kJ/mol, which is more typical for kinetically controlled processes. However, according to the literature [47,48,49,50,51,52], a high E_a_ value is not always allowed to make the final decision on the process nature.

The reaction order with respect to HNO_3_ concentration, Fe (III) ions concentration and amount of FeS_2_ additive were calculated using the graphical method (Table 4). The fractional order of reaction with respect to Fe (III) ions concentration and amount of FeS_2_ additive at Cu_12_As_4_S_13_, CuFeS_2_ and ZnS treatment suggests that the nature of the processes is diffusion controlled. At the same time, the reaction order with respect to HNO_3_ concentration at Cu_12_As_4_S_13_, CuFeS_2_ and ZnS treatment is more typical for chemical reaction control. The latter could be a result of aggressive impact on the S° layer that allows it to overcome the effect of passivation.

As a result, the research data were generalized and the general kinetic equations were established separately for Cu_12_As_4_S_13_, CuFeS_2_ and ZnS treatment, which consider the influence of temperature, concentration of reagents and duration of the experiments. As it shown in Figure 12, the relationship between the equations 1 – 3(1 – X)^2/3^ + 2·(1 – X) and C_HNO3_·C_Fe(III)_·C_FeS2_·exp[–E_a_/(R·T)]·τ·10^3^ for all experimental data was established, and the data points were evenly distributed along straight lines with a high R^2^.

The kinetic equations for treatment of Cu_12_As_4_S_13_, CuFeS_2_ and ZnS can be written as follows (11)–(13), respectively:Cu_12_As_4_S_13_: 1 − 3(1 − X) ^2/3^ + 2(1 − X) = 38820C_HNO3_^1.2^C_Fe(III)_^0.34^C_FeS2_^0.47^e^−28858/RT^t(11)
CuFeS_2_: 1 − 3(1 − X) ^2/3^ + 2(1 − X) = 74070C_HNO3_^1.42^C_Fe(III)_^0.82^C_FeS2_^0.69^e^−33708/RT^t(12)
ZnS:1 − 3(1 − X) ^2/3^ + 2 (1 − X) = 4.2C_HNO3_^1.52^C_Fe(III)_^0.62^C_FeS2_^0.59^e^−53723/RT^t(13)

Thus, the processes of sulfide minerals dissolution under investigated conditions are limited by internal diffusion [53]. The assessment was based on the obtained E_a_ values, orders of the reactions with respect to the reactants, SCM equations fitting and SEM-EDS investigation of the samples. Pyrite was proved as an effective catalytic surface for the reduction of nitrate ions and iron (III) with empirical order less than 1.

## 6. Conclusions

The current work was undertaken to deepen the understanding of the nature of the dissolution process for sulfides Cu_12_As_4_S_13_, CuFeS_2_ and ZnS in HNO_3_ media with application FeS_2_ and Fe (III) ions as catalysts.

It was observed that HNO_3_ concentration and temperature have the most significant influence on the conversion degree of Cu_12_As_4_S_13_, CuFeS_2_ and ZnS. The values of E_a_ were calculated as 28.8, 33.7 and 53.7 kJ/mol, respectively for Cu_12_As_4_S_13_, CuFeS_2_ and ZnS.

SEM-EDS scanning of the solid residues showed a presence of S° layer covering the surface of the minerals. The latter combined with E_a_ values and orders of the reactions with respect to the reactants obtained as well as SCM equations fitting allowed us to propose that the dissolution processes are of a diffusion nature.

It was additionally demonstrated that the presence of FeS_2_ in the system accelerates the conversion process due to galvanic coupling between minerals.

The results obtained can be used in predicting hydrometallurgical processes for sulfide materials such as copper–arsenic ores and concentrates treatment in HNO_3_ media.

Further detailed kinetic studies on the dissolution of sulfide minerals in HNO_3_ media such as Cu_3_AsS_4_, Cu_12_Sb_4_S_13_, Sb_2_S_3_, Cu_5_FeS_4_ are of great interest. Furthermore, the complex processing of the low-grade sulfide raw materials in HNO_3_ media is associated with the extraction of arsenic into the solution, which necessitates the following neutralization of nitrous gases as well as the arsenic utilization in the form of environmentally friendly compounds. These studies are of high relevance in terms of creating industrial hydrometallurgical technology.

## Figures and Tables

**Figure 1 materials-15-08149-f001:**
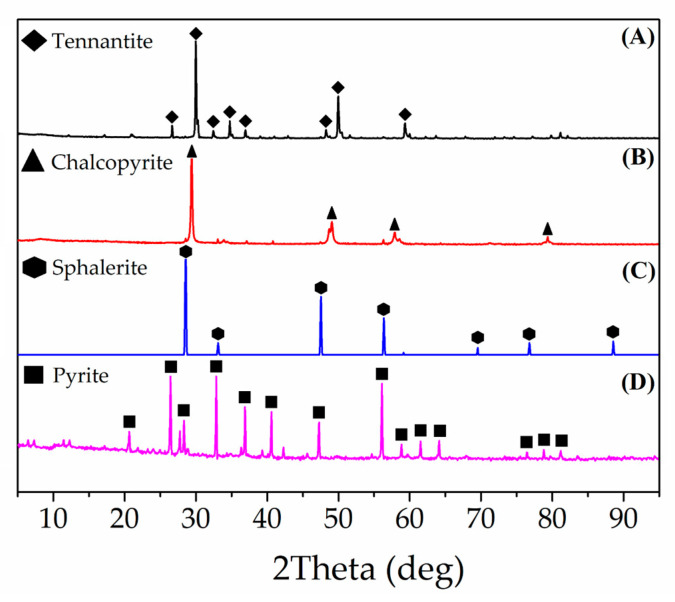
The XRD pattern of the minerals: Cu_12_As_4_S_13_ (**A**), CuFeS_2_ (**B**), ZnS (**C**) and FeS_2_ (**D**).

**Figure 2 materials-15-08149-f002:**
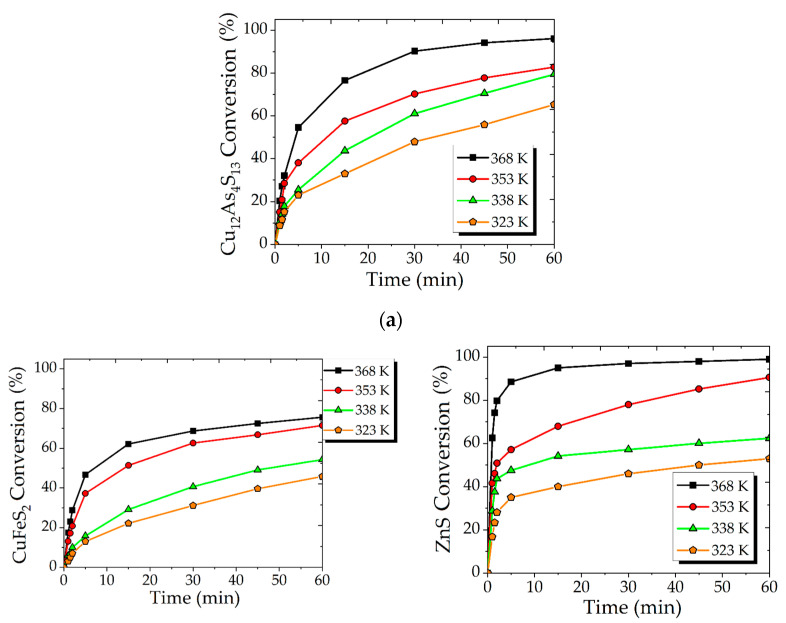
Effect of temperature on the conversion of Cu_12_As_4_S_13_ (**a**), CuFeS_2_ (**b**), ZnS (**c**). (300 rpm; 6 mol/L HNO_3_; 0.018 mol/L Fe (III); FeS_2_/synt. conc = 1/1).

**Figure 3 materials-15-08149-f003:**
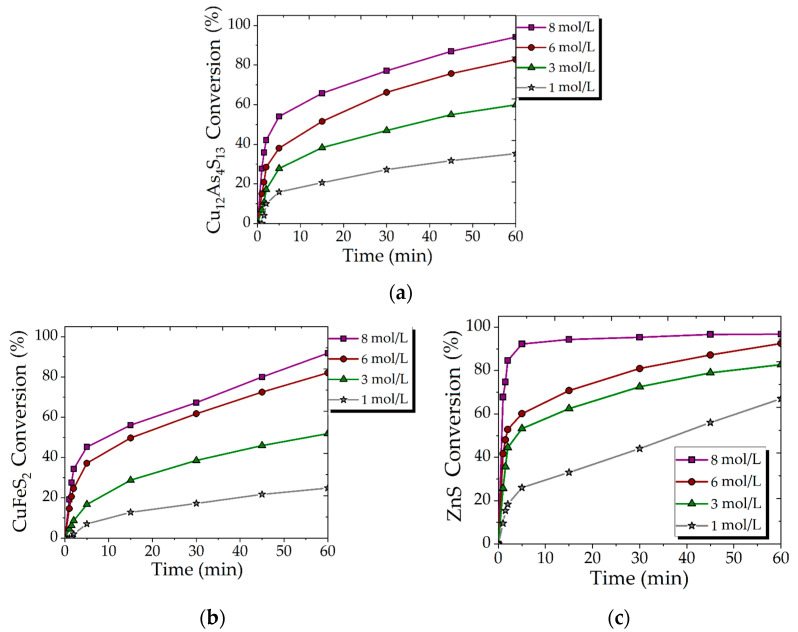
Effect of HNO_3_ concentration on the conversion of Cu_12_As_4_S_13_ (**a**), CuFeS_2_ (**b**) and ZnS (**c**) (300 rpm; 353 K; 0.018 mol/L Fe (III); FeS_2_/synt. conc = 1/1).

**Figure 4 materials-15-08149-f004:**
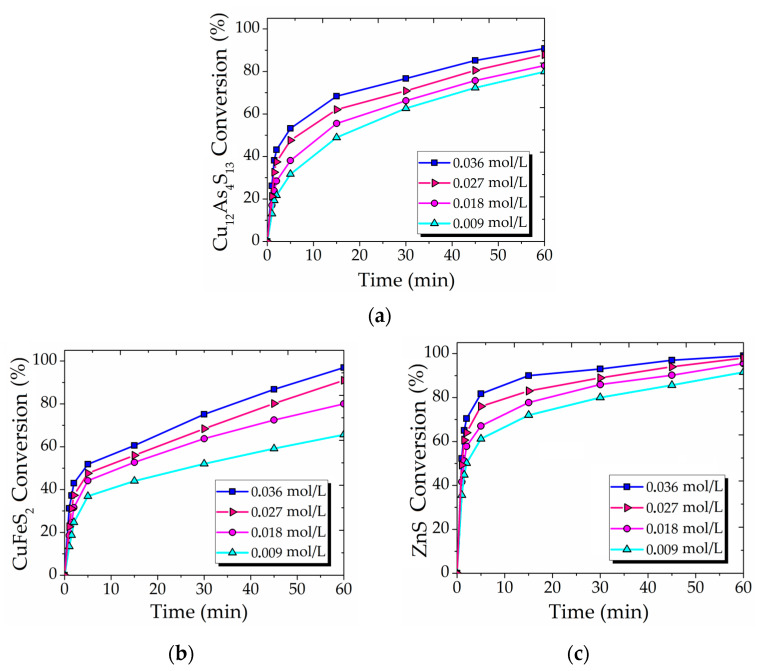
The effect of Fe (III) concentration on the conversion of Cu_12_As_4_S_13_ (**a**), CuFeS_2_ (**b**) and ZnS (**c**) (300 rpm; 353 K; 6 mol/L HNO_3_; FeS_2_/synt. conc = 1/1).

**Figure 5 materials-15-08149-f005:**
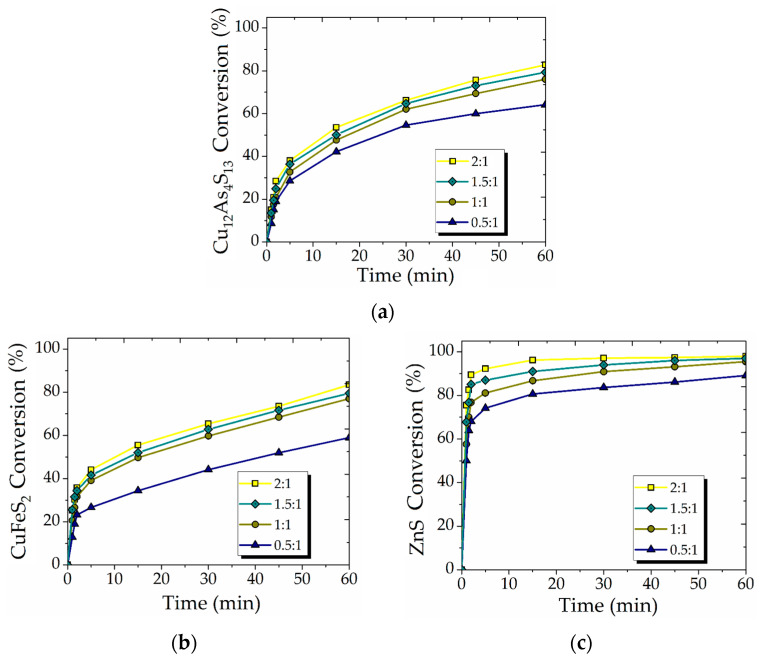
The effect of FeS_2_ additive on the conversion of Cu_12_As_4_S_13_ (**a**), CuFeS_2_ (**b**) and ZnS (**c**) (300 rpm; 353 K; 6 mol/L HNO_3_; 0.018 mol/L Fe (III)).

**Figure 6 materials-15-08149-f006:**
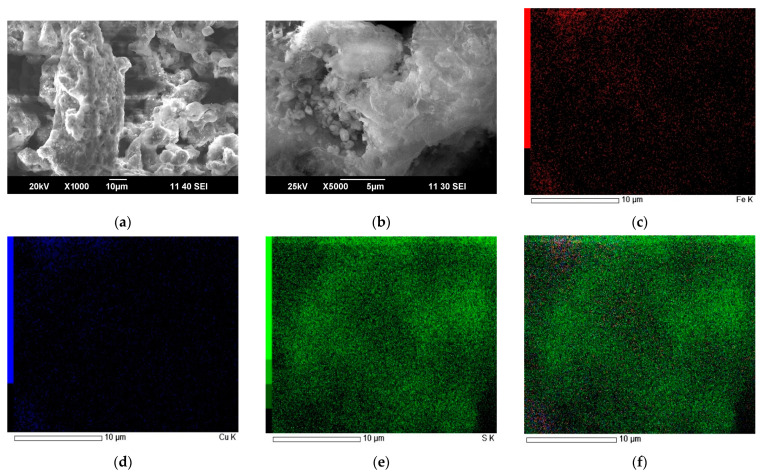
SEM images of synthetic concentrate-leaching residue (**a**,**b**) and EDX-mapping of the residue for Fe (**c**), Cu (**d**), S (**e**) and combined (**f**). (300 rpm; 353 K; 6 mol/L H_2_SO_4_; 0.018 mol/L Fe (III); 60 min; no FeS_2_ additive).

**Figure 7 materials-15-08149-f007:**
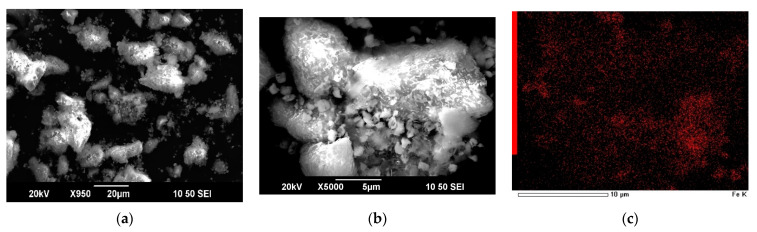
SEM images of synthetic concentrate-leaching residue (**a**,**b**) and EDX-mapping of the residue for Fe (**c**), Cu (**d**), S (**e**) and combined (**f**). (300 rpm; 353 K; 6 mol/L H_2_SO_4_; 0.018 mol/L Fe (III); 60 min; FeS_2_/synt. conc = 1/1).

**Figure 8 materials-15-08149-f008:**
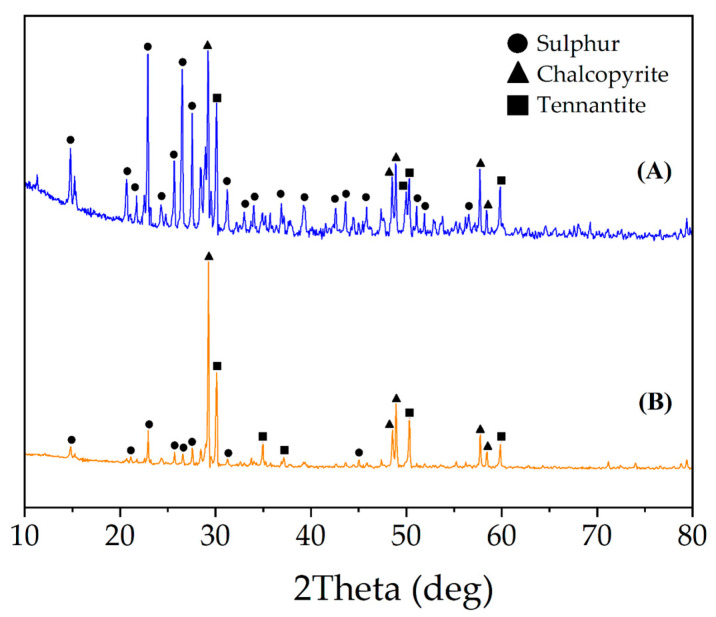
XRD patterns after synthetic concentrate dissolution in HNO_3_ solution without (**A**) and with (**B**) the additive of FeS_2_. (300 rpm; 353 K; 6 mol/L H_2_SO_4_; 0.018 mol/L Fe (III); 60 min).

**Figure 9 materials-15-08149-f009:**
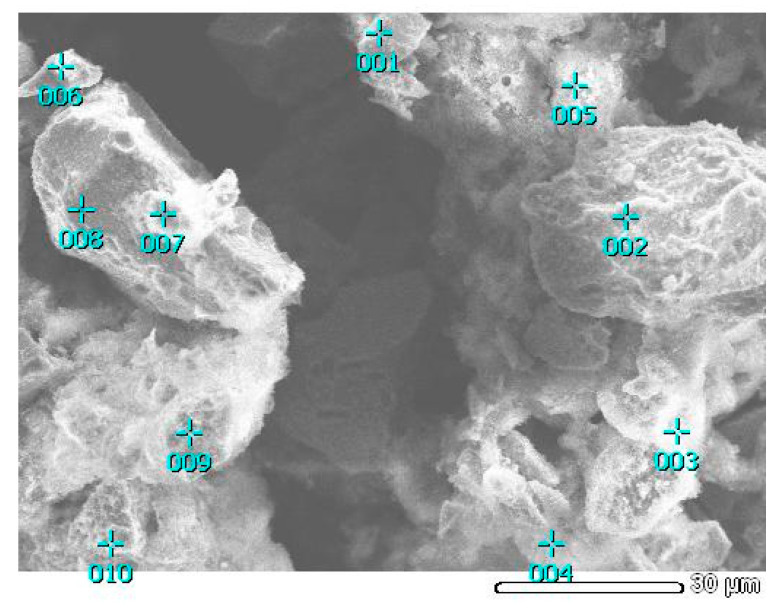
SEM images of the solid residue after synthetic concentrate dissolution in HNO_3_ solution. (300 rpm; 353 K; 6 mol/L H_2_SO_4_; 0.018 mol/L Fe (III); 15 min).

**Figure 10 materials-15-08149-f010:**
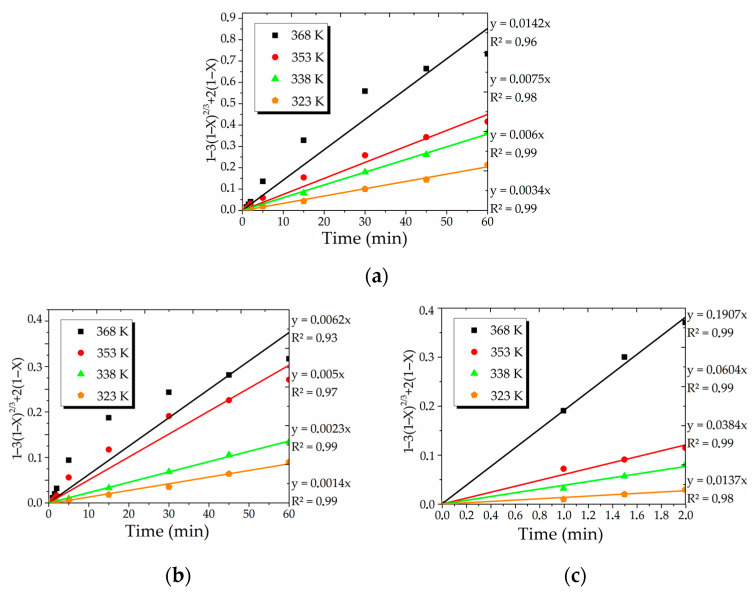
The linear relationship between 1 – 3(1 – X)^2/3^ + 2(1 – X) = k·τ and treatment time of Cu_12_As_4_S_13_ (**a**), CuFeS_2_ (**b**) and ZnS (**c**) at various temperatures.

**Figure 11 materials-15-08149-f011:**
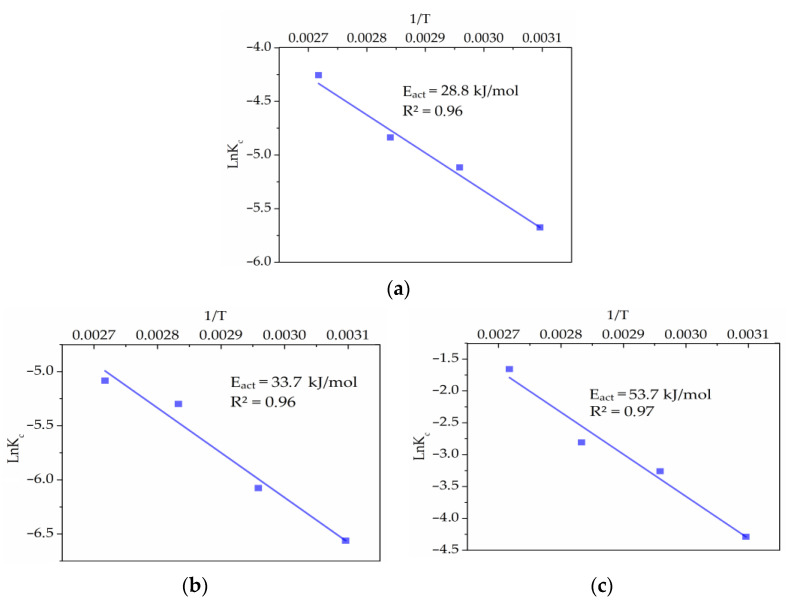
Arrhenius plots for Cu_12_As_4_S_13_ (**a**), CuFeS_2_ (**b**) and ZnS (**c**).

**Figure 12 materials-15-08149-f012:**
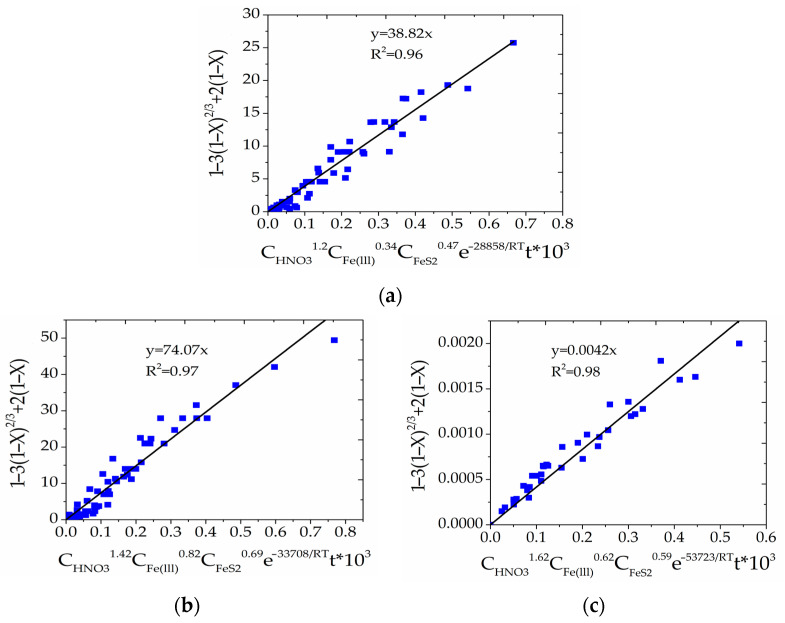
Relationship between SCM equation and C_HNO3_·C_Fe(III)_·C_FeS2_·exp[−E_a_/(R·T)]·τ·10^3^ for the treatment of Cu_12_As_4_S_13_ (**a**), CuFeS_2_(**b**) and ZnS (**c**).

**Table 1 materials-15-08149-t001:** Chemical composition of the synthetic concentrate (%).

Cu	Fe	S	As	Zn
41.8	7.15	30.2	13.2	7.45

**Table 2 materials-15-08149-t002:** Normalized EDX analysis results.

Element	Fe	Cu	As	S_sulfide_	S^0^	Total
Point 001	40.2	4.2	1.1	54.5	6.8	100.0
Point 002	43.8	2.3	0.8	53.1	2.0	100.0
Point 003	11.3	22.1	4.6	62.0	42.7	100.0
Point 004	23.4	21.7	1.6	53.3	24.3	100.0
Point 005	7.6	14.3	3.3	74.8	61.5	100.0
Point 006	29.4	11.9	2.3	56.2	19.1	100.0
Point 007	44.4	2.1	0.9	52.6	0.5	100.0
Point 008	45.3	1.5	0.3	52.9	0.6	100.0
Point 009	9.1	23.1	8.6	59.2	36.8	100.0
Point 010	9.5	30.4	10.4	49.7	24.4	100.0

**Table 3 materials-15-08149-t003:** Typical SCM kinetic equations applied for systems with spherical particles.

№	Limiting Stage	Formula
1	Diffusion through the product layer (sp)	1 − 3(1 − X) ^2/3^ + 2(1 − X)
2	Surface chemical reaction (sp)	1 − (1 − X)^1/3^

**Table 4 materials-15-08149-t004:** The reaction orders with respect to HNO_3_ concentration, Fe (III) ions concentration and the amount of FeS_2_ additive at Cu_12_As_4_S_13_, CuFeS_2_ and ZnS treatment.

Cu_12_As_4_S_13_	CuFeS_2_	ZnS
HNO_3_ concentration		
1.2	1.4	1.6
Fe(III) ions concentration		
0.34	0.82	0.62
Amount of FeS_2_		
0.47	0.69	0.59

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
