# Peer review of "Synthetic Sulfide Concentrate Dissolution Kinetics in HNO3 Media"

_materials, 2022, doi:10.3390/ma15228149_

Round 1
Reviewer 1 Report
The Authors have reported the dissolution studies of naturally occurring three important sulfides ores with HNO3. They have described the various effects of dissolution like concentrations of acids, timing, temperature and external metal atoms. Finally, they have established a model to correlate these studies theoretically.
I believe manuscript is well written and should be published. But before accepting, I have some minor comments and suggestion which may improve the readability of the manuscript.
(1) Authors have dissolved the compound in HNO3 but for the timing and temperature variation study, they did not mention the concentration. I thing concentration of HNO3 should be mentioned for all the studies.
(2) If the dissolution kinetics is different for ZnS, then why there is a same kinetic model for the SCM equation.
(3) In the temperature variation studies, ZnS has a faster rate of dissolution (Figure 2c) in high temperature but when the activation energy has been calculated it has the highest activation energy for dissolution (Figure 11c). I believe that creates confusion. Author should clarify the matter.
(4) Author should fit the dissolution kinetics experimental data and talk about the reaction order.
Author Response
Question 1.
Authors have dissolved the compound in HNO3 but for the timing and temperature variation study, they did not mention the concentration. I think concentration of HNO3 should be mentioned for all the studies.
Answer:
Thanks for the comment. HNO3 and Fe(III) concentrations, stirring speed, FeS2 to synthetic concentrate mass ratio and temperature as the constant experimental conditions are given below the Figures 2–5. In our opinion, this is the most concise option to display the experimental conditions. Please let us know if you think another option is more appropriate.
Question 2.
If the dissolution kinetics is different for ZnS, then why there is a same kinetic model for the SCM equation.
Answer:
Thanks for the comment. There was a mistake in the abstract. Information that the ZnS conversion is characterized as kinetic-controlled was taken from another study while language editing. The mistake has been corrected.
Question 3.
In the temperature variation studies, ZnS has a faster rate of dissolution (Figure 2c) in high temperature but when the activation energy has been calculated it has the highest activation energy for dissolution (Figure 11c). I believe that creates confusion. Author should clarify the matter.
Answer:
Indeed, the Ea value for ZnS is more typical for kinetically controlled processes. However, the kinetic analysis considering other indicators showed that the diffusion equation is the most appropriate. Literary overview (below) showed that this phenomenon occurs in hydrometallurgical processes. Appropriate explanations have been added into the manuscript.
Baba, A.A., Adekola, F.A. Hydrometallurgical processing of a Nigerian sphalerite in hydrochloric acid: Characterization and dissolution kinetics. Hydrometallurgy 2010, 101, 69–75.
Barlett, R. Upgrading copper concentrate by hydrothermal converting chalcopyrite to digenite. Metallurgical Transactions B, Process Metallurgy 1992, 241–248.
Fuentes, G., Viñals, J., Herreros, O. Hydrothermal purification and enrichment of Chilean copper concentrates Part 1: The behavior of bornite, covellite and pyrite. Hydrometallurgy 2009, 104–112.
Gok, O, Anderson, C.G., Cicekli, G., Cocen, E.I. Leaching kinetics of copper from chalcopyrite concentrate in nitrous-sulfuric acid. Physicochem. Probl. Mi. 2014, 50, 399–413
Kritskii, A., Celep, O., Yazici, E., Deveci, H., Naboichenko, S. Hydrothermal treatment of sphalerite and pyrite particles with CuSO4 solution. Minerals Engineering 2022, 107507
Viñals, J., Fuentes, G., Hernández, M. Transformation of sphalerite particles into copper sulfide particles by hydrothermal treatment with Cu(II) ions. Hydrometallurgy 2004, 177–187.
Question 4.
Author should fit the dissolution kinetics experimental data and talk about the reaction order.
Answer:
Thanks for the comment. Appropriate explanations have been added into the manuscript.
Reviewer 2 Report
1) It’s not clear to me that this is a novel idea,
2) The Abstract and the conclusion should be more quantitative,
3) Add nomenclature to the manuscript,
4) Recommendation and perspectives are missing in the conclusion section,
5) Are your all results presensented in table 1, 2, and 4 the average of different runs?
6) Compare and thoroughly discuss your results with the literature,
7) Literature review should be updated (arround 95 % references are from 2016 and previous years),
8) This article needs to be formatted carefully and the language should be re-approved by a native speaker/Grammarly checked. It’s so difficult to read through the article.
Author Response
Dear Reviewers!
Thanks for all of your comments and suggestions. We have corrected the manuscript and answered your questions:
Question 1.
It’s not clear to me that this is a novel idea
Answer:
Thanks for the comment. Indeed, a simple idea to subject sulfides for dissolution on HNO3 media is not new. In this work, a detailed kinetic study on HNO3 dissolution of Cu12As4S13, CuFeS2 and ZnS was carried out to deepen the understanding of the reaction mechanism. Moreover, kinetic studies on HNO3 dissolution of Cu12As4S13 with application of the modern approaches in kinetic analysis are completely absent in the open sources. Our team will be grateful if you share references on the research topic, which are beyond the scope of the references list present in the manuscript.
Question 2.
The Abstract and the conclusion should be more quantitative
Answer:
Thanks for the comment. The abstract and conclusion sections have been edited.
Question 3.
Add nomenclature to the manuscript
Answer:
Thanks for the comment. The terminology used is considered as standart and common for manuscripts on hydrometallurgy in MDPI journals. Please check out below-mentioned articles. Please let us know what exactly is causing the confusion.
Dizer O.A., Rogozhnikov D.A., Karimov K.A., Kuzas E.A., Sunrsuv A.Yu. Nitric Acid Dissolution of Tennantite, Chalcopyrite and Sphalerite in the Presence of Fe (III) Ions and FeS2. Materials 2022, â„– 1545.
Kritskii A.V., Naboichenko S.S. Hydrothermal Treatment of Arsenopyrite Particles with CuSO4 Solution. Materials 2021, â„– 7472.
Kritskii A., Celep O., Yazici E., Deveci H., Naboichenko, S. Hydrothermal treatment of sphalerite and pyrite particles with CuSO4 solution. Minerals Engineering 2022, â„– 107507.
Kritskii A., Naboichenko S., Karimov K., Agarwal V., Lundström M. Hydrothermal pretreatment of chalcopyrite concentrate with copper sulfate solution. Hydrometallurgy 2020, â„– 105478.
Rogozhnikov D.A., Shoppert A.A., Dizer O.A., Karimov K.A., Rusalev R.E. Leaching Kinetics of Sulfides from Refractory Gold Concentrates by Nitric Acid. Metals 2019, â„– 465.
Zheng X., Li Sh., Liu B., Zhang L., Ma A. A Study on the Mechanism and Kinetics of Ultrasound-Enhanced Sulfuric Acid Leaching for Zinc Extraction from Zinc Oxide Dust. Materials 2022, â„– 5969.
Question 4.
Recommendation and perspectives are missing in the conclusion section
Answer:
Thanks for the comment. The conclusion section has been edited.
Question 5.
Are your all results presensented in table 1, 2, and 4 the average of different runs?
Answer:
Thanks for the comment. Yes, analysis methods and experimental runs involved parallel measurements for proving the results.
Question 6.
Compare and thoroughly discuss your results with the literature?
Answer:
Thanks for the comment. The "Results and discussion" section have been edited. Relevant references have been added.
Question 7.
Literature review should be updated (arround 95 % references are from 2016 and previous years)
Answer:
Thanks for the comment. Our team tried our best to expand the reference list.
Question 8.
This article needs to be formatted carefully and the language should be re-approved by a native speaker/Grammarly checked. It’s so difficult to read through the article.
Answer:
Thanks for the comment. Indeed, some spelling errors were found. The manuscript has been re-edited for the quality of the English language. Please provide examples, which cause confusion. This manuscript, like many previous ones (listed below), was translated into English by a colleague holding ILETS and Cambridge English certificates.
Kritskii A., Celep O., Yazici E., Deveci H., Naboichenko, S. Hydrothermal treatment of sphalerite and pyrite particles with CuSO4 solution. Minerals Engineering 2022, â„– 107507.
Kritskii A.V., Naboichenko S.S. Hydrothermal Treatment of Arsenopyrite Particles with CuSO4 Solution. Materials 2021, â„– 7472.
Kritskii A., Naboichenko S., Karimov K., Agarwal V., Lundström M. Hydrothermal pretreatment of chalcopyrite concentrate with copper sulfate solution. Hydrometallurgy 2020, â„– 105478.
Round 2
Reviewer 2 Report
No response to my last comment, I keep the recommendation: Reconsider after major revision,
1/Add nomenclature to the manuscript
2/Recommendation and perspectives are missing in the conclusion section
3/Literature review should be updated (around 95 % references are from 2016 and previous years)
Author Response
Question 1.
Add nomenclature to the manuscript
Answer:
Thanks for the comment. Nomenclature has been added.
Question 2.
Recommendation and perspectives are missing in the conclusion section
Answer:
Thanks for the comment. Recommendation and perspectives have been added.
Question 3.
Literature review should be updated (around 95 % references are from 2016 and previous years).
Answer:
Thanks for the comment. Literature review has been updated.